# Exploring the Role of Neuroplasticity in Development, Aging, and Neurodegeneration

**DOI:** 10.3390/brainsci13121610

**Published:** 2023-11-21

**Authors:** Patrícia Marzola, Thayza Melzer, Eloisa Pavesi, Joana Gil-Mohapel, Patricia S. Brocardo

**Affiliations:** 1Department of Morphological Sciences and Graduate Neuroscience Program, Center of Biological Sciences, Federal University of Santa Catarina, Florianopolis 88040-900, SC, Brazil; patriciarmarzola@gmail.com (P.M.); melzer.th@gmail.com (T.M.); eloisapavesi@gmail.com (E.P.); 2Division of Medical Sciences, University of Victoria, Victoria, BC V8P 5C2, Canada; 3Island Medical Program, Faculty of Medicine, University of British Columbia, Victoria, BC V8P 5C2, Canada

**Keywords:** aging, cognitive function, lifestyle interventions, neurodegeneration, neurodevelopment, neuroplasticity

## Abstract

Neuroplasticity refers to the ability of the brain to reorganize and modify its neural connections in response to environmental stimuli, experience, learning, injury, and disease processes. It encompasses a range of mechanisms, including changes in synaptic strength and connectivity, the formation of new synapses, alterations in the structure and function of neurons, and the generation of new neurons. Neuroplasticity plays a crucial role in developing and maintaining brain function, including learning and memory, as well as in recovery from brain injury and adaptation to environmental changes. In this review, we explore the vast potential of neuroplasticity in various aspects of brain function across the lifespan and in the context of disease. Changes in the aging brain and the significance of neuroplasticity in maintaining cognitive function later in life will also be reviewed. Finally, we will discuss common mechanisms associated with age-related neurodegenerative processes (including protein aggregation and accumulation, mitochondrial dysfunction, oxidative stress, and neuroinflammation) and how these processes can be mitigated, at least partially, by non-invasive and non-pharmacologic lifestyle interventions aimed at promoting and harnessing neuroplasticity.

## 1. Introduction

The concept of neuroplasticity was first introduced by William James in 1890, and a few decades later, Jerzy Konorski coined the term “neural plasticity” [1,2]. Neuroplasticity refers to changes in brain structure and function throughout the lifespan. Neuroplasticity enables the brain to change and adapt to intrinsic or extrinsic stimuli by reorganizing its structure, function, or connections, resulting in physiological and morphological modifications. This dynamic process allows us to adjust to different experiences and circumstances and plays a significant role in learning, memory, and recovery from brain injuries [3].

Due to the multifaceted nature of neuroplasticity, different types of plasticity can impact brain structure and function [4,5]. Structural neuroplasticity refers to changes in the physical structures of neurons and neural networks, including the number, shape, strength, and connectivity of synapses [6], thus enabling the brain to adapt to changing environments and experiences. Numerous studies have indicated that structural plasticity occurs during development and continues into adulthood [7,8,9]. On the other hand, functional neuroplasticity refers to changes in neural network properties that involve efficiency, strength, and synchrony changes of synapses. Functional plasticity occurs rapidly, affecting various cognitive and behavioral processes relating to attention, memory, and perception [5,10].

One well-studied example of structural neuroplasticity is adult neurogenesis, the process by which new neurons are generated in the adult brain. This process occurs primarily in the subventricular zone (SVZ) that lines the lateral ventricles and in the dentate gyrus of the hippocampus, a brain region essential for learning and memory [9,11,12,13]. Several studies have shown that increased physical activity, exposure to enriched environments, and certain drugs can enhance neurogenesis and improve learning and memory [8,14,15,16]. Another example of structural neuroplasticity is dendritic spine remodeling, the process by which dendritic spines change in size, shape, and number in response to experience. Animal studies have shown that dendritic spine remodeling also plays a crucial role in learning and memory [17,18].

Functional neuroplasticity is thought to underlie memory formation, skill acquisition, and recovery from injury. An example of functional neuroplasticity is long-term potentiation (LTP), the persistent strengthening of synapses in response to repeated stimulation. LTP is thought to be a key mechanism underlying learning and memory [19]. Conversely, long-term depression (LTD) is the persistent weakening of synapses and also plays a role in learning and memory [20,21]. Another example of functional neuroplasticity is cortical reorganization, the process by which the brain’s sensory maps can change in response to experience or injury. Learning new abilities results in changes to functional connectivity among brain areas involved with motor control, sensory processing, and attention. For example, blind individuals can have enhanced sensory processing in other modalities, such as touch and hearing, due to cortical reorganization [22].

In the perinatal and early childhood periods, the brain undergoes rapid and extensive growth and development, during which plasticity is particularly high. Studies have shown that this period is characterized by a heightened sensitivity to environmental input, which facilitates the formation of new neural connections [23]. In contrast, plasticity in later stages of the lifespan is more tightly regulated and context-dependent. Changes in neural activity, environmental factors, and behavioral outcomes can trigger the release of specific neurotransmitters, enabling changes in neural connections only under contextual conditions that facilitate plasticity [24]. Moreover, recent studies have suggested that the regulation of plasticity in the mature brain occurs as a continuum, with different levels of plasticity occurring under different conditions [25]. These findings suggest that plasticity is a dynamic process that can be modulated and affected by various factors, including age, experience, and environmental conditions. Understanding these factors can aid in developing effective strategies to harness the power of neuroplasticity and minimize its negative effects, leading to better treatments and outcomes for various neurological and neurodegenerative conditions.

## 2. Neuroplasticity

Neuroplasticity, also known as brain plasticity or neural plasticity, is the biological capacity of the brain to adapt physiologically or even alter its anatomical structure in response to stimuli or damage [26]. This ability is central to learning, memory, injury recovery, and adaptation to environmental changes [27].

### 2.1. Structural Neuroplasticity

Structural neuroplasticity refers to physical changes to neural circuits, including the growth of new dendritic spines, axonal sprouting, and even neurogenesis. In particular, neurogenesis refers to generating new functional neurons, a multifaceted and tightly regulated process involving the proliferation, differentiation, and integration of new neurons from neural precursor cells. Each stage is characterized by the activation and presence of distinct transcriptional factors and markers [12,28,29,30]. Structural neuroplasticity is essential to rewiring the brain and has implications for recovery after brain injury, neurodevelopment, and adaptations to sensory input alterations throughout life [31].

#### 2.1.1. Developmental Neurogenesis and Synaptogenesis

The development of the central nervous system (CNS) begins in the early weeks following fertilization, shortly after formation of the three embryonic germ layers, ectoderm, mesoderm, and endoderm, a crucial phase during embryonic development. The CNS derives from the differentiation of multipotent cells present in the ectoderm through the formation of the neural plate in the dorsal region of the embryo. This neural plate will fold its crest in the craniocaudal and rostrocaudal directions, forming the neural tube at embryonic day 30 in the human [32]. The closure of the neural tube marks the beginning of rapid brain enlargement from two groups of cells: neural stem cells (NSCs), which are multipotent cells that can give rise to various types of neural cells, including neurons, astrocytes, and oligodendrocytes, and express markers such as Sox 2 and Nestin; and neural precursor cells (NPCs), which are immediate descendants of neural stem cells, are committed to a neuronal fate and express markers such as Pax6, Dlx2, and Tbr2 [30,32]. Between weeks 4 and 5 of human embryonic development begins a phase known as interkinetic nuclear migration of NSCs and NPCs, which results in the symmetric division along the ventricular edge. This early proliferation leads to an exponential increase in the pool of progenitor cells that contribute to the expansion of surface area and thickness of the ventricular zone [30,33,34]. Around gestational week 5 (human embryonic day 42), the NPCs located in the ventricular zone, referred to as radial glial cells, begin to switch from symmetric to asymmetric cell division, generating one daughter cell that remains in the ventricular zone as a radial glial cell and a postmitotic neuron, marking the beginning of neurogenesis itself [33]. The process of differentiation of NPCs into neurons involves the successive expression of specific transcription factors and proteins, including Neurogenin (Ngn) and Mash 1 (Ascl1), doublecortin (DCX), βIII-tubulin (Tuj1), and finally NeuN, a marker of mature neurons [34]. Of note, new neurons also need to migrate to their appropriate locations within the brain. This process involves the support of radial glial cells and Cajal–Retzius cells, which create pathways such as the reelin pathway to aid migrating neurons in reaching their final destination [35].

A final step in the neurogenic process involves establishing functional connections between the newly generated neurons to form neural circuits. This process relies on synaptic plasticity or synaptogenesis. Synaptogenesis begins approximately in human gestational week 27 and continues to occur after birth. During the postnatal period, synapses are produced rapidly; by age two, the number of synapses is estimated to be twice the number in the adult brain. Indeed, synaptogenesis is an incredibly dynamic process in the human cerebral cortex in infancy and childhood [27], with the postnatal period being marked by enhanced experience-dependent sensitivity to sensory information. In addition, synaptic strength and efficacy alterations occur during the development [27], allowing the developing brain to adapt to environmental stimuli. This “critical period” is not sustained into adulthood, though, thus restricting the ability to indiscriminately store new sensory information [36]. Indeed, the number of synapses falls over the subsequent years and into adolescence through a process referred to as synaptic pruning, through which necessary synapses are preserved and redundant ones are eliminated [37]. These findings support the idea of “windows of opportunity” that enable the construction and consolidation of experience-dependent structural and functional brain connections during the neurodevelopment period [38] and explain why children can readily acquire new languages (and other skills). At the same time, this ability requires much more effort and attention later in life [39].

Of note, in addition to being influenced by environmental stimuli, the various stages of developmental neurogenesis and synaptogenesis are tightly regulated by numerous intrinsic factors [32,34], including transcription factors, growth factors (such as brain-derived neurotrophic factor (BDNF) and insulin-like growth factor (IGF)), cell adhesion molecules (such as N-cadherin, which aids in cell migration and synaptic connection formation), and signaling molecules (such as Notch, Sonic Hedgehog (SHH), Wnt, and fibroblast growth factor (FGF)).

#### 2.1.2. Adult Neurogenesis

Although the original evidence in support of adult neurogenesis dates back to the 1960s with the pioneering work of the American neuroscientist Joseph Altman [28], it was only in the mid-1990s that this phenomenon became generally accepted [40] following the seminal work by Eriksson et al. (1998), who demonstrated the incorporation of bromodeoxyuridine (BrdU; a nucleotide analog) into the DNA of newly generated neurons in the human hippocampal dentate gyrus [41]. Since then, numerous studies have confirmed that neural stem cells are indeed present in juvenile and adult brains [42] and that neurogenesis continues to occur in select regions of the adult mammalian brain [14,43], the most important and widely studied being the subgranular zone (SGZ) of the hippocampal dentate gyrus [28] and the subventricular zone (SVZ) of the lateral ventricles [44]. Newborn neurons have also been described in other brain regions, referred to as “noncanonical” neurogenic areas [12], including the hypothalamus [45,46], neocortex [47,48], amygdala [49], cerebellum [50,51], and striatum [42,52]. Of note, adult neurogenesis is thought to have functional significance. For example, adult hippocampal neurogenesis is involved in several emotional and cognitive functions, including spatial learning, memory, pattern separation, and mood regulation [11,53,54,55].

Similarly to developmental neurogenesis, adult neurogenesis is also modulated by several intrinsic and extrinsic factors, including trophic support [56,57], epigenetic factors [58], physical activity [16], stress [59], environmental enrichment [14], and pharmacological interventions, such as antidepressants [60,61,62]. On the other hand, a reduction or dysregulation of the neurogenic processes may contribute, at least in part, to alterations in cognition and emotional resilience seen with normal aging [63,64,65], as well as with various psychiatric and neurodegenerative conditions, including addiction [66], autism spectrum disorder [67,68], major depressive disorder [69], Huntington’s disease (HD) [70], and Alzheimer’s disease (AD) [71].

### 2.2. Functional Neuroplasticity

In contrast to structural neuroplasticity, functional neuroplasticity refers to changes in the functional organization of neural circuits. As mentioned in this review, synaptogenesis represents a structural aspect of plasticity but also has functional implications. The formation of new synapses or the strengthening of existing ones can enhance the communication and transmission of signals among neurons, leading to functional changes in neuronal circuits. These adaptations include two forms of synaptic plasticity, long-term potentiation (LTP) and long-term depression (LTD), through which the strength of synaptic connections between neurons can change in response to different patterns of neuronal activity, thus contributing to memory formation, skill acquisition, and habituation [72]. According to this hypothesis, learning or experiencing something new can strengthen synaptic connections. This, in turn, increases the efficiency of synaptic neurotransmission, ultimately aiding in memory consolidation and information recall [73,74].

The term LTP was first introduced in 1973 by Bliss and Lomo [75]. The authors demonstrated the long-lasting increase of synaptic strength in the dentate gyrus of the hippocampus following high-frequency stimulation. Later, in 1993, Bliss and Collingridge expanded on these findings and discussed the potential implications of LTP in the context of a cellular memory model [19]. LTP is one of the most well-studied mechanisms underlying neuroplasticity and is a specific cellular and synaptic process in which the strength of a synapse is increased, resulting in more efficient transmission of signals between the presynaptic and the postsynaptic neurons. Mechanistically, LTP involves changes that can last for an extended period (days, weeks, or even years) in the synaptic structure, such as an increase in the size and shape of postsynaptic dendritic spines as well as the increase in the area of postsynaptic density (PSD) and the number of neurotransmitter receptors of the postsynaptic membrane in response to the calcium-dependent activation of N-methyl-D-aspartate (NMDA) receptors [76]. LTP is often described using the principle of Hebbian plasticity, which states that synapses that are repeatedly active at the same time tend to strengthen their connections. This principle aligns with the idea that the brain adapts to experiences and reinforces the neural pathways associated with those experiences [77].

On the other hand, LTD is the process of decreasing synaptic strength, resulting in a less efficient transmission of signals in response to the depolarization of the postsynaptic neuron for an extended period. LTD helps maintain synaptic connections’ overall balance and efficiency, which is essential for synaptic homeostasis [21]. Both LTP and LTD are critical for the adaptative capabilities of the CNS and allow neural circuits to adjust their connections and synaptic strength in response to experiences, cognitive function, memory consolidation, and habituation [72,73].

In summary, functional and structural plasticity play an important role in brain function, and changes in neuroplasticity may be associated with diseases and disorders of the CNS. It is worth noting that neuroplasticity is most robust during development, but it persists throughout life. This fact has significant implications for understanding brain function, recovery from brain injury, and potentially treating neurological and psychiatric disorders.

## 3. Neurodevelopment and Neuroplasticity 

The brain is arguably the most complex organ in the human body, and its development is a continuous process that starts early in gestation and continues throughout adulthood [78]. Neurodevelopment is a highly orchestrated process involving the proliferation, migration, differentiation, and maturation of neurons and the formation and refinement of synaptic connections between them [79]. Normal neurodevelopment is essential for the proper functioning of the nervous system, and any disruption in this process can lead to a wide range of neurodevelopmental and neurological disorders, such as autism spectrum disorder (ASD), attention-deficit/hyperactivity disorder (ADHD), schizophrenia, and epilepsy [80]. The causes of these disorders are complex and multifactorial, involving genetic and environmental factors that can affect brain development and function [7]. Studies have shown that neuroplasticity involves many aspects of brain development, including forming and refining neural connections during early development and acquiring new skills and abilities throughout life [81,82]. In addition, neuroplasticity plays a crucial role in recovery from brain injury and stroke and in treating neurological and psychiatric disorders, such as depression and anxiety [83,84].

On the other hand, neuroplasticity disruptions can adversely affect neurodevelopment. Exposure to toxic substances during development, such as alcohol or drugs, can impair neuroplasticity and disrupt normal brain development [85,86]. Similarly, experiences with chronic stress or trauma can impair neuroplasticity and lead to permanent changes in brain structure and function [87]. Understanding the role of neuroplasticity in normal brain development is critical for identifying and addressing factors that can disrupt this process and lead to neurodevelopmental and neurological disorders. This section discusses the stages of normal neurodevelopment and the factors influencing this process.

### 3.1. Prenatal Stage (from Conception until Birth)

Neurodevelopment is a critical process that begins approximately two to three weeks after conception with the formation of the neural tube [88]. Neurulation involves the folding and fusion of the lateral ends of the neural plate and is a crucial stage in the formation of the brain and spinal cord. Abnormalities during neurulation can result in neural tube defects, such as spina bifida [89]. Following neurulation, neurogenesis and neuronal migration occur, which are also critical processes in neurodevelopment. Neurogenesis occurs primarily during the first trimester of pregnancy and in specific brain regions, such as the ventricular zone [29]. Neuronal migration occurs during the second trimester of pregnancy. It is essential for proper brain development, allowing neurons to form correct connections with other neurons and establish functional neural circuits [90].

During neurogenesis and neural migration, environmental factors such as stress, nutrition, alcohol exposure, and sensory input can influence the rate and direction of these processes. For example, exposure to stress hormones during gestation has been shown to alter the timing of neurogenesis and the migration of new neurons, leading to changes in brain structure and function [91]. 

Another significant event during embryonic development is the division of the neural tube into three primary brain vesicles, which give rise to the different structures of the brain [92]. Microglia migrate into the developing brain during the early stages of gestation, around 4–5 weeks after fertilization, and establish the pool of resident immune cells [93]. Gliogenesis, which produces region- and subtype-specific glia, begins at 22 weeks of gestation and continues throughout adulthood [94]. Glial cells also aid in the myelination of neurons at approximately 32 weeks of pregnancy, a process that continues after birth and into adulthood [95]. At 18 weeks of gestation, the excess of cells is eliminated through apoptosis, a form of programmed cell death, resulting in the refinement of cell populations and ensuring proper development and synaptic connectivity in the mature brain [96]. Synaptogenesis begins in utero at approximately 27 weeks of gestation but predominantly occurs after birth, coinciding with the growth of dendrites and axons and axonal myelination [38]. The pruning of excess synapses and dendritic processes continues after birth, a necessary process for proper neural network formation, with disruptions of this step being linked to various neurological disorders [97]. By the end of the prenatal period, major fiber pathways, such as the thalamocortical pathway, have been established [34]. These critical stages of brain development are essential for a healthy and functional brain. Conversely, disruptions in fetal brain development have been linked to an increased risk of psychiatric disorders, such as autism and schizophrenia [98]. The complexity of these processes highlights the importance of proper prenatal care, including good nutrition and avoiding harmful substances such as alcohol and drugs, which can adversely affect brain development [85,99].

The developmental origins of the health and disease hypothesis (DOHaD) propose that environmental exposure during early life (particularly during the prenatal period) can permanently influence the long-term development of disease. The initial studies addressing this relationship observed the association between gestational malnutrition and the phenotypes of the offspring, as well as the risk of developing metabolic diseases such as obesity, diabetes, and cardiovascular diseases later in life. In accordance with this, later studies have identified epigenetic modifications in fetal DNA as a response to environmental stimuli, which can permanently alter protein expression and phenotypes in the offspring. Of note, some of the environmental factors causing epigenetic modifications besides maternal nutrition include smoking, maternal stress, and infection [100,101]. 

During prenatal development, genetic information plays a crucial role in orchestrating events that determine the formation and refinement of neural connections. A complex sequence of guidance molecules instructs newly born cells on what type of neurons to become and where to go, while genetically determined intrinsic neural activity instructs the refinement of axonal projections, which are guided by molecular cues to their approximate target area [102]. While these events are critical for proper brain development, prenatal disruptions can negatively affect neuroplasticity. For example, prenatal exposure to teratogenic factors such as alcohol can interfere with glutamatergic and GABAergic neurotransmitter function, destabilizing previously tentative synapses [103,104]. This interference can lead to structurally different yet functionally viable circuits. Studies have also shown that neuroplastic changes during the prenatal period can have long-lasting effects on brain development and behavior. For example, maternal stress during pregnancy has been linked to altered brain development and increased risk for behavioral disorders such as ADHD and ASD in the offspring, negatively impacting cognitive and emotional outcomes later in life and increasing the risk for mental health conditions in adulthood [105].

Interestingly, through non-invasive techniques such as fetal magnetoencephalography (fMEG) that record neural responses to external stimuli like music, speech, and touch [106], it has been shown that exposure to music during the prenatal period can enhance the development of the auditory system and improve cognitive function later in life [107]. In addition, studies have shown that the fetus can recognize and respond to familiar voices, including the mother’s voice. This recognition can be attributed to the development of the auditory system, which is functional by the 16th week of gestation [108].

### 3.2. Infancy and Childhood

After birth, the human brain undergoes refinement and reorganization, particularly during sensitive and critical periods known as “windows of brain plasticity,” which are most pronounced in early childhood but continue into adolescence and adulthood [38]. During this stage, the brain undergoes an extraordinary growth spurt, with neurons forming connections at an astonishing rate. As the brain triples in size during the first two years of life, it builds an immense network of neural circuits that enables the processing of sensory information and the development of higher-order cognitive functions [109]. Studies have shown that early experiences and environmental factors are essential in shaping the developing brain. For example, language development begins early in life. It involves a complex interplay of genetic and environmental factors [110] and is strongly influenced by early exposure to language and sounds [111,112].

Another critical aspect of neurodevelopment in childhood (as well as adolescence; see below) is the development of executive skills such as attention, working memory, and self-control. These skills are important for academic achievement, social functioning, and well-being [113]. The development of executive functioning skills is influenced by several factors, including genetics, environmental factors, and experiences [114]. Brain neuroplasticity during infancy is also essential for developing social and emotional skills. Infants develop the ability to recognize and respond to emotional cues and to form secure bonds with caregivers. These skills, critical for social and emotional development, are shaped by early experiences and essential for later well-being [115]. The growth and development of the brain during this period lays the foundation for later cognitive, social, and emotional abilities. Therefore, providing infants and children with a nurturing and stimulating environment that promotes and supports optimal brain development is essential.

Windows of opportunity are necessary for developing new skills; however, they leave the brain vulnerable to the harmful effects of the environment. In the same way that environmental stimuli lead to necessary and expected neuronal remodeling while learning a new skill that will be important throughout life, adverse situations can potentially result in maladaptive changes. The World Health Organization (WHO) has coined the term early stress to refer to adverse situations that occur during childhood, from birth to 18 years of age. Early stress can include coping with conditions such as living in extreme poverty, violence, abuse (physical, sexual, or psychological), neglect, and grief, among other examples. According to the WHO, around one billion children and adolescents are exposed to some type of early stress every year [116,117]. When a child experiences a form of early stress, the hypothalamic–pituitary–adrenal (HPA) axis is activated, resulting in the release of cortisol, which in turn activates the sympathetic nervous system, thus triggering the fight-or-flight response as a defense mechanism [118].

However, when the stress response is chronic and exacerbated, it leads to adverse effects, including deregulating the HPA axis. Indeed, the sustained release of high levels of cortisol into the blood plasma can result in the disruption of the feedback loop that regulates the HPA axis, causing resistance to cortisol and ultimately resulting in the damage of various brain regions, including the hippocampus [119,120]. Indeed, when in excess, cortisol (or corticosterone in rodents) causes damage to dendritic arborization, the morphology of dendritic spines, and the synaptic integrity of hippocampal neurons. Male C57BL/6N mice that were subjected to an environmental stress protocol (insufficient sawdust in the housing box) between postnatal days 2 and 9 showed decreased arborization and lower density of dendritic spines in pyramidal neurons in the CA3 region of the hippocampus [121]. Furthermore, exposure to early stress induces decreased neurogenesis in the dentate gyrus of the hippocampus in C57 adult mice [122].

Notably, the impact of hyperactivation of the HPA axis on brain morphology and behavior (including the development of self-destructive behaviors, less tolerance to adverse everyday situations, greater vulnerability to substance abuse, and difficulties in interpersonal relationships) is well documented in the literature [123,124]. Indeed, reductions in adult neurogenesis, dendritic arborization, and glucocorticoid receptor density changes have been observed in the hippocampus of adults who experienced early stress [124,125]. Furthermore, exposure to early stress has also been associated with decreased gray matter and increased psychiatric disorders, including anxiety and depression [126]. Furthermore, studies have demonstrated the relationship between the experience of early stress in childhood and the development of alcohol dependence later in life [127]. A survey of 3592 adults on their drinking habits and history of early stress showed that the average age at which alcohol intake began was lower in people with traumatic experiences. While most participants said they drink to socialize and feel good, around 10% said they use alcohol to “deal with problems and stress” [128]. In addition, Pilowsky and collaborators (2009) demonstrated an association between traumatic events occurring in childhood and adolescence with a greater frequency of heavy episodic drinking and an early onset of alcohol consumption, with data highlighting that experiencing two or more traumatic events early in life increased the propensity for alcohol dependence in adulthood. The chronicity and severity of these episodes were also associated with the risk of relapse in female patients who were undergoing treatment for cocaine addiction [129,130].

### 3.3. Adolescence

As children grow into adolescence, their brains mature, accompanied by significant cognitive, social, and emotional development. Indeed, neuroimaging studies have shown that the adolescent brain undergoes considerable changes in the prefrontal cortex, which is responsible for higher-order cognitive functions, such as decision making, impulse control, attention, and working memory [131]. The prefrontal cortex is also involved in social and emotional processing, and its development during adolescence is crucial for acquiring social and emotional skills, including navigating complex social relationships, empathizing with others, and regulating emotions. Indeed, the prefrontal cortex undergoes significant changes in its structural and functional connectivity during adolescence, and these changes have been related to improvements in social cognition and emotion regulation [132,133]. Moreover, recent studies have highlighted the role of the social brain network, which includes regions such as the medial prefrontal cortex, the temporoparietal junction, and the amygdala, in social and emotional processing during adolescence [134,135]. These brain regions are involved in social cognition, empathy, and emotional regulation, and their development during adolescence is crucial for social and emotional competence.

Understanding the neurobiological changes during adolescence is crucial for promoting healthy brain development and preventing mental health conditions during this critical period of life. One of the most significant changes during adolescence is synaptic pruning, which involves the elimination of unnecessary synapses and neural connections. This process makes the brain more efficient by reducing neural noise and enhancing information processing [136]. Synaptic pruning occurs mainly in the prefrontal cortex but also affects other brain regions, such as the hippocampus and amygdala [37].

### 3.4. Adulthood

In adulthood, the rate of neurodevelopment slows down significantly. However, the brain retains the capacity to form new neurons and connections and adapt to new experiences throughout life. One mechanism through which the brain can continue to adapt is adult neurogenesis, which is thought to play an important role in learning and memory, as well as in mood regulation and the stress response [11,53,54,55].

In addition to the hippocampus and the SVZ, additional brain regions have also emerged as sites where adult neurogenesis can take place [12]. Indeed, animal studies have delineated neurogenic loci encompassing the hypothalamus [137], striatum [138,139,140], substantia nigra (SN) [141], cerebral cortex [142], and amygdala [143]. Evidence indicates that the genesis of neurons in these newly identified neurogenic areas is attributable to the migration of NSPCs, typically originating from the SVZ [140,143,144,145,146,147]. Concurrently, some studies have challenged this notion and proposed the existence of endogenous pools of NSPCs within these regions, capable of local replication and integration into neuronal circuits [137,138,148,149].

The hypothalamus, one of the major regulatory centers in the brain, controls various homeostatic processes, and hypothalamic neural stem cells (htNSCs) have been shown to interfere with these processes. Indeed, the hypothalamic neurogenesis is thought to influence metabolism and fat storage, as evidenced by multiple studies on the impacts of a high-fat diet (HFD) in mice [46,150,151]. Furthermore, neurogenesis within the hypothalamus is also thought to contribute to behavioral and sexual functions, as elucidated in studies by Bernstein et al. (1993) [152], Fowler et al. (2002) [153], and Cheng et al. (2004) [154]. Additionally, emerging research suggests that neurogenesis in the hypothalamus undergoes alterations during aging [155], prompting investigations into the potential implications of age-related changes in hypothalamic neurogenesis on overall physiological homeostasis and cognitive functions [156].

Of note, deficits in adult hippocampal neurogenesis (as well as other forms of structural and functional plasticity) have been implicated not only in normal aging [63], but also in various psychiatric [69] and neurodegenerative [70,71] conditions. Conversely, classic antidepressants such as monoamine oxidase inhibitors (MAOIs), tricyclic antidepressants, and selective serotonin reuptake inhibitors (SSRIs) have been shown to possess pro-neurogenic properties, and these are thought to mediate, at least in part, their antidepressant effects [60,61,157,158]. More recently, ketamine, an anesthetic with antidepressant properties, was also shown to increase adult hippocampal neurogenesis in rodents [159]. Other studies have explored the potential of environmental enrichment (such as exposing animals to a stimulating environment with toys and social interactions) [8,14,160] and physical exercise [160,161,162] in promoting adult hippocampal neurogenesis. These findings highlight the exciting potential of pharmacologic and non-pharmacologic interventions in promoting adult hippocampal neurogenesis and potentially improving human brain health and cognitive function.

### 3.5. Prenatal Factors That Impact Neurodevelopment and Neuroplasticity

Several prenatal factors can significantly impact neurodevelopment, and multiple lines of research have identified maternal nutrition, exposure to toxins (e.g., alcohol and illicit drugs), and infection as crucial determinants. Insufficient nutrition during pregnancy can lead to low birth weight and impaired cognitive development [163]. Moreover, maternal nutrition has been associated in preclinical and clinical research with altered neuropsychiatric outcomes in the offspring [164,165]. Of note, insulin has been identified as a critical modulator of neuronal network development during the early phases of life [166]. Studies have shown an association between impaired insulin signaling in the hippocampus of adolescent and adult offspring of obese mice with impairments in hippocampal gene expression, neurogenesis, and synaptic plasticity [167,168,169,170]. Furthermore, a recent study in rodents also suggested that an appropriate maternal diet, especially fiber-rich, could regulate and reverse these neurocognitive alterations [170]. One possible mechanism underlying the effects of a maternal high-fat diet on these neurodevelopment outcomes is related to altered Notch 1 signaling activation, which in turn is thought to inhibit the proliferation and differentiation of neural progenitors [171]. In addition, maternal excess salt intake has also been associated with changes in brain development and neural plasticity in rodents, particularly concerning synaptic transmission and neuroplasticity in the hippocampus [172].

Exposure to toxins, including alcohol, illicit drugs, and heavy metals such as lead, significantly impacts the developing brain, potentially resulting in brain damage and developmental delays [173]. These toxic substances can enter the body through various sources, including environmental pollution, contaminated food and water, and maternal substance use. Alcohol exposure during pregnancy has been linked to fetal alcohol spectrum disorders (FASDs), a range of neurodevelopmental and behavioral problems that can result in lifelong disabilities and neurocognitive abnormalities [85,174].

Consumption of cannabis (marijuana) during the prenatal period has also been shown to affect neurodevelopmental processes. Indeed, in utero exposure to Δ9-tetrahydrocannabinol was associated with behavioral alterations in adolescent rats, including impairment in aversive limbic memory, decreased instrumental learning, and increased alcohol consumption [175]. Similarly, exposure to illicit drugs, including stimulants (such as cocaine and methamphetamine) and opioids (such as heroin), can cause a range of adverse effects, including congenital disabilities, low birth weight, and developmental delays [176]. Exposure to heavy metals can also cause irreversible damage to the developing brain, leading to a range of cognitive, behavioral, and developmental problems [177]. Maternal infections during pregnancy can also have severe consequences for fetal development. For example, rubella infection during pregnancy can lead to congenital malformations, including hearing loss and intellectual disabilities [178]. Although the association between COVID-19 and congenital anomalies in babies conceived and born during the pandemic is still unclear due to the lack of knowledge on fetal and perinatal complications following COVID-19 infection, some studies have shown an increase in the rate of CNS congenital anomalies during the pandemic [179,180,181]. The long-term neuroplastic changes caused by COVID-19 infections are still unclear and require further investigation.

### 3.6. Postnatal Factors That Impact Neurodevelopment and Neuroplasticity

After birth, several postnatal factors can significantly influence neurodevelopment and neuroplasticity. Nutrition, social interaction, and environmental factors can all affect brain development during the postnatal period. Nutritional status is a critical determinant of brain growth and development, particularly during the first 1000 days of life [182]. In rodents, a high-fat diet during the postnatal period can negatively impact cognition and synaptic plasticity and promote neuroinflammation and microglial activation [183]. For example, ingestion of sweeteners, such as aspartame and sucralose, can induce changes in behavior and neuroplasticity, including decreases in hippocampal neurogenesis and BDNF levels in rats in a sex-dependent manner [184]. In humans, breastfeeding has been associated with improved metabolic and neurocognitive health outcomes in infants [185], effects that are thought to be mediated, at least in part, by the unique properties and lipid composition of maternal milk [186]. In addition, several studies have shown that inadequate nutrition during infancy and early childhood can lead to long-term cognitive deficits, reduced academic achievement, and behavioral problems [187,188]. In contrast, adequate nutrition during this period can promote optimal brain development, including improved cognitive and behavioral outcomes [189,190].

As explained above (see Section 3.2), the early years of life are a period of significant neuroplasticity, and experiences during this time can shape the structure and function of the brain [191]. Social interaction and early childhood experiences, such as language exposure, are critical for optimal brain development. Studies have shown that children who experience high-quality early care and education, including language-rich environments, have better cognitive, social–emotional, and academic outcomes than those who do not [192,193]. Conversely, lack of stimulation, social deprivation, and neglect can have long-lasting adverse effects on brain development, including reduced cortical thickness and gray matter volume [194] and increased risk of cognitive deficits and developmental delays [195].

Life experience and environmental enrichment positively modulate behavior and neuroplasticity during critical periods and throughout life [36,196]. Environmental enrichment during childhood, adolescence, and adulthood has been shown to promote neurogenesis and affect the pattern of monosynaptic inputs in animal models [14,160,197]. Additionally, environmental enrichment has been shown to improve spatial learning performance and neuroplasticity while increasing hippocampal volume and BDNF levels in various animal models [160,198]. In humans, evaluating the impact of environmental enrichment on brain neuroplasticity and function is much more complex due to the number of potential confounding variables. However, a recent randomized control trial has assessed the effects of environment and neurodevelopment and concluded that interventions that can reduce poverty could promote changes in children’s brain function and the development of higher-order cognitive skills. This study found that infant neuroplasticity was positively correlated with economic status, which in turn is known to impact numerous socio-economic factors that can influence a child’s well-being, including household income and expenses, type and amount of work of the mother, parenting behavior, and overall family wellbeing and stress [199]. Social interactions can also be considered a form of environmental enrichment known to modulate neuroplasticity. In particular, maternal contact has been shown to impact neurodevelopment and to have long-lasting effects on behavior. Appropriate maternal care during the first postnatal week can promote life-long stress resilience in rodents [200]. 

In conclusion, neuroplasticity is a critical determinant of brain growth and development not only prenatally but also during the postnatal period. Several prenatal factors, such as maternal nutrition, exposure to toxins, and infection, can potentially result in or contribute to the offspring’s cognitive, behavioral, and developmental problems. On the other hand, an appropriate maternal diet, especially one rich in fiber, can help prevent neurocognitive alterations caused by a maternal high-fat diet. The early years of postnatal life are also particularly significant for brain development, as experiences during this time can shape the structure and function of the brain. Adequate nutrition, language-rich environments, high-quality early care, education, and rewarding social interactions and stimulation are critical for optimal brain development during this period. Environmental enrichment during childhood and adolescence can increase neuroplasticity and improve cognitive outcomes. In contrast, exposure to toxins, social deprivation, neglect, and harmful environmental factors can have a long-lasting negative impact on brain development, potentially leading to long-term cognitive deficits and behavioral and psychological disturbances.

### 3.7. Sex Hormones and Neuroplasticity

Sex hormones are now known to have widespread actions in both the male and female brains, through mechanisms thought to involve both genomic and nongenomic receptors. Indeed, many neural and behavioral functions are affected by sex hormones such as estrogens, including mood, cognitive function, blood pressure regulation, motor coordination, pain, and opioid sensitivity [201]. Moreover, sex-specific differences have been reported with regards to hippocampal-dependent cognition and neurogenesis, suggesting that sex hormones are involved in these processes. Indeed, estrogens have been shown to modulate certain forms of spatial and contextual memory, as well as different forms of neuroplasticity including neurogenesis, primarily in the adult female hippocampus [202,203].

Peripheral sex steroid hormones, including estrogens, progesterone, testosterone, and other androgens, are able to cross the blood–brain barrier and reach the brain. Furthermore, hippocampal neurons are capable of synthesizing sex steroids de novo from cholesterol, since neural cells express all the enzymes required for the synthesis of estradiol and testosterone, the end products of sex steroidogenesis [204,205,206,207,208,209]. Regarding 17β-estradiol (E2) in particular, its synthesis in hippocampal neurons is homeostatically controlled by Ca^2^**^+^** transients and is regulated by the release of gonadotropin-releasing hormone (GnRH) [210]. Indeed, release of GnRH from GnRH-positive neurons in the hippocampus is thought to regulate the local synthesis of sex steroids in a sex-dependent manner and thus contribute to the sexual differentiation of hippocampal neurons during the perinatal period [210,211]. Of note, a GnRH-induced increase in estradiol synthesis appears to provide a link between the hypothalamus and the hippocampus, and this may underlie, at least in part, estrous cyclicity of spine density in the female hippocampus [206,210]. Furthermore, sex hormones can initiate gene transcription and activate signaling cascades by utilizing genomic and non-genomic [201,212] mechanisms that play a key role in coordinating various physiological and pathological neuroplasticity-related events, such as formation or remodeling of dendritic spines, neurogenesis, synaptogenesis, and myelination modulation [213]. For example, the study by Lu et al. in 2019, employing a forebrain-neuron-specific aromatase knock-out mouse model, provided compelling genetic evidence of the involvement of neuron-derived E2 in modulating AKT-ERK and CREB-BDNF signaling cascades. This study established that neuron-derived E2 is essential for normal expression of LTP and other forms of synaptic plasticity, as well as cognitive function in both male and female brains [214]. However, it has also been proposed that whereas E2 appears to be essential to maintaining synaptic transmission and synaptic connectivity in the female hippocampus, dihydrotestosterone appears to be crucial for synaptic transmission and synaptic connectivity in the male hippocampus [207,210,211]. As the expression of sex hormones varies throughout the lifespan, its effects on neuroplasticity during distinct periods of development, adulthood, and aging must also be considered. Along these lines, strategies aimed at restoring and/or maintaining normal hormone levels in the brain throughout the lifespan such as physical exercise are thought to be beneficial in promoting brain health in general and neuroplasticity in particular [215].

## 4. Aging, Neurodegeneration, and Neuroplasticity

### 4.1. Physiological Aging

Aging is a natural process affecting various physiological systems, including the immune, cardiovascular, musculoskeletal, and nervous systems [216]. According to the United Nations, the proportion of people over 60 is projected to reach 25% by 2050 [217,218], reflecting a global trend towards population aging. As the population ages worldwide, the incidence of age-related diseases and the associated healthcare costs also increase.

The World Health Organization (WHO) defines healthy aging as the process of optimizing opportunities for physical, mental, and social well-being to enable older individuals to maintain their functional abilities that allow them to participate actively in society [219]. This definition encompasses not only the absence of disease but also emphasizes the importance of maintaining functional capacity and engagement in activities that contribute to a fulfilling life. For older adults, healthy aging involves healthy lifestyle habits and behaviors, including healthy nutrition, regular physical activity, and avoiding smoking, excessive drinking, and illicit drugs. 

Aging increases the risk of developing chronic diseases such as cardiovascular disease, diabetes, and cancer. For example, type 2 diabetes is more prevalent in older adults, with approximately one in four adults over the age of 65 being affected by this metabolic disorder [220]. Cancer incidence also increases with age, with approximately 60% of all cancer cases occurring in adults over the age of 65 [221]. Moreover, ischemic heart disease, stroke, and chronic obstructive pulmonary disease (COPD) are common causes of mortality in the elderly population [222]. Additionally, aging is characterized by a decline in immune function linked to the accumulation of senescent cells, which secrete pro-inflammatory cytokines and contribute to chronic inflammation, ultimately resulting in tissue damage [223]. Indeed, aging is associated with cellular senescence, a state of irreversible growth arrest, which can be induced by various stressors, such as oxidative stress, resulting in the damage of cellular proteins, lipids, and DNA [224], as well as telomere shortening [225].

One of the critical differences between physiological and pathological brain aging relates to cognitive reserve and the maintenance of cognition in physiological aging versus the presence of notable cognitive decline in pathological aging. Indeed, the intrinsic capacity of the nervous system to regenerate and compensate for neuronal loss (i.e., the “brain reserve”) may counteract neurodegeneration and its consequences [226]. Within this scenario, environmental (e.g., lifestyle) and genetic factors influence the degree of resiliency to aging, thus influencing how much the intrinsic “brain reserve” can help prevent age-related neuronal damage and loss. Cognitive reserve depends on how efficiently the system can tap into the brain reserve [227]. In animal studies, several intrinsic and extrinsic factors, including sex, body weight, physical activity, sleep duration, and anxiety and stress, can all affect the cognitive reserve and the onset of cognitive impairments with aging [228]. 

On the other hand, regulation of neuroinflammation and oxidative stress, as well as maintenance of calcium homeostasis, can promote cellular resilience and neuronal circuit adaptation and ultimately increase cognitive reserve [229,230,231]. Nevertheless, the aging brain does experience significant structural and functional changes, particularly affecting the hippocampus and cerebral cortex [63,232]. Depending on the cognitive reserve of the brain, alterations in neurons and their connectivity may decrease cognitive function [63] and increase the risk of developing age-related neurological disorders such as AD and Parkinson’s disease (PD) [233,234,235,236]. 

In addition to affecting neuronal structure and function, aging has also been associated with astrocytic dysfunction and subsequent synaptic disturbances. Indeed, in old animals, deficiencies in astrocytic glutamate uptake have been related to synaptic plasticity impairments and age-related cognitive decline. Furthermore, astrocytes’ role in maintaining antioxidant defenses may also be compromised by aging, which can further contribute to neurodegeneration [237].

In addition, microglia (immune cells of the brain) and oligodendrocytes (myelinating cells of the CNS) are also affected by aging. Indeed, an age-related reduction in both white matter volume, myelinization, and microglia activation has been observed in the aged brain, with an impairment in microglia activation being associated with decreased neuroprotection and a compromised process of synapse elimination [237]. Neuroinflammation is a well-established contributor to age-related cognitive impairments and hippocampal neurogenesis deficits. The production of inflammatory mediators, including cytokines, interleukins, and neurotrophins, as well as the activation of glia and other immune cells, can have a direct impact on synaptic plasticity and neurogenesis and with that exacerbate the processes associated not only with normal aging but also with neurodegenerative disorders [238,239].

One of the most notable changes associated with aging is mitochondrial dysfunction. With age, mitochondria become less efficient at producing adenosine triphosphate (ATP) and more prone to producing toxic reactive oxygen species (ROS) [216], which in turn can cause neuronal dysfunction and, ultimately, cell death. Dysregulation of mitochondrial function contributes to the pathogenesis of age-related diseases, including neurodegenerative disorders. Indeed, mitochondrial dysfunction has been implicated in AD and PD [240]. The accumulation of mitochondrial DNA mutations has been observed in aged mice, and this has been linked to increased oxidative damage [241]. Intracellular Ca^2+^ dysregulation has also been implicated in age-related cognitive decline [242,243]. Indeed, an increase in intracellular Ca^2+^ levels in dorsal hippocampus CA1 pyramidal neurons has been shown to contribute to memory impairment in aged animals [244]. Furthermore, reduced NMDA receptor activation and LTP can impair Ca^2+^-dependent synaptic plasticity in old animals [245].

Interventions targeting the various physiological systems affected by aging may increase the lifespan and improve the quality of life of older individuals. For example, physical exercise has been shown to improve cardiovascular function, increase neuroplasticity, and promote the production of anti-inflammatory cytokines, all of which can help mitigate the adverse effects of aging on the body and the brain [246,247]. Furthermore, promoting healthy lifestyle habits such as maintaining a nutritious diet, engaging in regular physical exercise, and avoiding smoking and excessive alcohol consumption can also help minimize the impact of aging on individuals and society [63,162]. Indeed, maintaining functional capacity and improving well-being can enhance the quality of life for older individuals and drastically reduce healthcare costs.

### 4.2. Neurodegeneration

Neurodegeneration is characterized by the progressive loss of neuronal structure and function, leading to irreversible neuronal damage and cell death [248]. Neurodegeneration underlies the development of several neurodegenerative diseases, including AD, PD, as well as HD, and amyotrophic lateral sclerosis (ALS) [249,250,251]. Neurodegenerative diseases affect millions worldwide, with AD and PD being the most common neurodegenerative disorders. In the United States, as many as 6.2 million people may have AD, as reported by the Alzheimer’s Disease Association in 2022 [252]. Similarly, nearly a million Americans are living with PD, according to the Parkinson’s Foundation [253]. The incidence of these disorders is expected to triple by 2050, highlighting the need for effective prevention and treatment strategies [254].

Multiple factors are known to contribute to the neurodegenerative processes that culminate in neuronal damage and, ultimately, cell death [238,255]. For example, protein misfolding, aggregation, and deposition have long been recognized as neuropathological hallmarks common to many neurodegenerative disorders, including AD, PD, HD, and ALS [250,256,257,258]. Moreover, mitochondrial dysfunction (which can result from genetic mutations, exposure to environmental toxins, as well as physiological aging) [259], ROS generation and accumulation, and a consequent increase in oxidative stress [250,260,261] have all been shown to play a role in the pathogenesis of several neurodegenerative disorders, including AD, PD, and HD. In addition, neuroinflammation has also been implicated in the development of neurodegenerative diseases [262,263,264]. Other factors contributing to aging and neurodegeneration (and consequent neuroplasticity impairment) include genetic and environmental factors [236].

#### 4.2.1. Protein Aggregation in Neurodegeneration

Neurodegenerative diseases are characterized by the accumulation and aggregation of disease-specific proteins such as beta-amyloid and tau in AD, alpha-synuclein in PD, and mutant huntingtin in HD [265]. These proteins misfold and form toxic oligomers and fibrils that interfere with normal cellular functions, eventually leading to cell death. This process is known as protein aggregation. The formation of abnormal protein aggregates is believed to arise from disturbances in the proteostasis network. This tightly regulated system ensures proper protein folding, trafficking, and degradation under normal conditions [266]. Disruptions to the proteostasis network can result from genetic mutations, aging, environmental stimuli, or a combination of several factors. Such disruptions can trigger the accumulation of misfolded proteins, formation of protein aggregates, increased neuroinflammation and oxidative stress, and activation of apoptotic pathways, ultimately culminating in neuronal death.

The endoplasmic reticulum (ER) is a critical component of the proteostasis network and is involved in the folding and processing of proteins [267]. Disruptions to ER function can accumulate misfolded proteins and trigger ER stress, activating the unfolded protein response (UPR) [268]. The UPR signaling pathway activates adaptive pathways to improve protein folding and promote quality control mechanisms and degradative pathways [269]. Recent studies have shown that the UPR plays a role in the pathogenesis of neurodegenerative diseases. For example, in AD, the UPR is activated in response to the accumulation of beta-amyloid [270], while in PD, it is activated in response to the accumulation of alpha-synuclein [271]. However, if the UPR fails to restore proteostasis or if the accumulation of misfolded proteins exceeds the capacity of the cellular machinery to degrade them, the UPR can also activate apoptotic pathways, thus resulting in neuronal death [266]. 

#### 4.2.2. Mitochondrial Dysfunction and Oxidative Stress in Neurodegeneration

Oxidative stress and mutations in mitochondrial DNA contribute to aging and neurodegenerative diseases [272]. Several studies have shown that mitochondrial DNA mutations accumulate in the aging brain and are associated with cognitive decline [261,273,274]. Moreover, increased levels of ROS have been shown to induce mitochondrial DNA mutations and damage the mitochondrial respiratory chain, leading to mitochondrial dysfunction and neuronal death [275,276,277]. Furthermore, recent studies have shown that the crosstalk between oxidative stress and other pathological mechanisms, such as protein misfolding and inflammation, can further exacerbate mitochondrial damage and neuronal death [278].

An increasing number of disease-specific proteins have been found to interact with mitochondria. For example, mutant huntingtin has been shown to disrupt mitochondrial function and dynamics in HD [279]. Moreover, dysfunctional mitochondria may precipitate AD since beta-amyloid disrupts mitochondrial function and impairs energy production in brain cells [280,281]. PD is also associated with mitochondrial dysfunction, including impaired mitochondrial dynamics and oxidative stress [282], and oxidative stress has been linked to alpha-synuclein accumulation in dopaminergic neurons, a hallmark of the disease [283]. Together, these studies suggest that mitochondrial dysfunction is involved in the pathogenesis of multiple neurodegenerative disorders.

#### 4.2.3. Neuroinflammation in Neurodegeneration

Neuroinflammation is a complex process mediated by microglia and astrocytes and is thought to play a role in several neurodegenerative diseases [284]. Microglia can assume phagocytic phenotypes and release inflammatory cytokines in response to various stimuli, including protein aggregates and pathogens [285]. For example, protein aggregates, such as beta-amyloid in AD and alpha-synuclein in PD, can activate microglia and induce chronic neuroinflammation, leading to neuronal dysfunction and death [286,287]. Moreover, oligomeric aggregates of beta-amyloid, tau, and alpha-synuclein can initiate both glial and neuronal inflammation by activating several inflammatory pathways [286,288,289]. In agreement, a recent study has shown that induced peripheral inflammation can potentiate the adverse effects of alpha-synuclein oligomers by exacerbating neuroinflammation and cognitive deficits in a synucleinopathy mouse model [290].

In response to injury or inflammation, activated astrocytes (i.e., astrogliosis) can also contribute to neuroinflammation by releasing cytokines and chemokines, further activating microglia and perpetuating neuroinflammation [291]. In agreement, a recent study showed that activated microglia can promote the activation of neurotoxic reactive astrocytes (A1 type), resulting in both neuron and oligodendrocyte death. Of note, reactive A1 astrocytes have been reported in various neurodegenerative diseases, including AD, PD, HD, ALS, and multiple sclerosis [287]. In animal models of AD, reactive astrocytes were associated with atrophy of glial fibrillary acidic protein (GFAP)-positive cells and the presence of amyloid deposits in brain regions such as the hippocampus [292].

#### 4.2.4. Genetic and Environmental Factors in Neurodegeneration

Genetic and environmental factors can also contribute to neuroinflammation and neurodegeneration [293]. For example, mutations in the triggering receptor expressed on the myeloid cells 2 (TREM2) gene, expressed by microglia, have been linked to an increased risk of AD and other neurodegenerative diseases [294]. Moreover, genome-wide association studies have identified new genetic risk factors for AD and PD [295].

In addition, environmental factors such as exposure to toxins, infection, and traumatic brain injury (TBI) have also been shown to increase the risk of various neurodegenerative diseases [296,297,298]. For example, exposure to pesticides, heavy metals, and solvents has been linked to increased incidence of AD and PD [296]. Viral infections, such as herpes simplex virus type 1 and human herpes virus 6, have also been associated with the development of AD [297]. Furthermore, TBI has been linked to the development of chronic traumatic encephalopathy, a neurodegenerative disease commonly found in athletes and military veterans [298].

### 4.3. Correlation between Aging, Neurodegeneration, and Neuroplasticity

Several lines of evidence have supported a strong correlation between aging, neurodegeneration, and neuroplasticity. With age, both neurogenesis (i.e., the generation of new neurons) and synaptic plasticity (i.e., the ability of neurons to form new connections) decline [299]. These changes are thought to contribute to the development and progression of neurodegenerative processes by impairing the ability of the brain to compensate for the effects of physiological aging and/or incurred damage while maintaining normal function. On the other hand, engaging in activities that promote neuroplasticity, such as learning new skills or engaging in regular physical exercise, has been shown to help maintain cognitive function and slow cognitive decline in older adults [162,300]. These findings suggest that interventions designed to enhance neuroplasticity may slow or potentially reverse the effects of neurodegeneration in older adults.

The hippocampus and hippocampal neuroplasticity are particularly affected by the aging process. Age-related changes in the hippocampus, such as increased oxidative stress, neuroinflammation, altered gene expression, hormone imbalance, reduced neurogenesis, and impaired synaptic plasticity, have all been associated with cognitive decline [63]. Indeed, several studies have shown that aging is associated with decreased synaptic plasticity, including LTP and LTD, which are thought to underlie learning and memory. For example, hippocampal LTP was reduced in older adults, and this decrease was associated with poorer memory performance [301]. Of note, the decline in hippocampal LTP observed in aging rats was also shown to be associated with a decrease in the expression of estrogen receptors in the hippocampus [302]. Furthermore, aged animals were shown to have decreased synaptic plasticity in the projections from the entorhinal cortex to the dentate gyrus [303] and reduced neuronal excitability of CA1 pyramidal neurons [304]. Conversely, improving cyclic AMP response element-binding protein (CREB) signalizing enhanced cognitive performance in aged animals [305].

In addition, aging has also been associated with a decline in structural plasticity throughout the brain, including the hippocampus. Indeed, aging is associated with reduced dendritic arborization and length, spine density [306,307,308], as well as decreased synaptogenesis (i.e., formation of new synapses) [299]. Furthermore, adult neurogenesis also dramatically declines with age, with the proportion of neuronal stem cells that survive to become mature neurons being significantly reduced in the aged brain [309,310]. Of note, such decreases have been shown to impact learning strategies in aged mice [310]. In humans, neuroimaging studies have shown that aging is associated with a decrease in gray matter volume and cortical thickness, which may reflect a reduction in synaptic density [311].

In summary, the decline in neuroplasticity accompanying aging has important implications for cognitive function and the risk of developing neurodegenerative diseases. Understanding the mechanisms underlying the age-related decline in neuroplasticity is crucial for developing strategies to promote healthy aging and reduce the risk of age-related neurological disorders. To this end, non-invasive strategies aimed at improving neuroplasticity, including physical exercise, cognitive stimulation, social engagement, and dietary interventions, hold promise in halting the course of neurodegeneration in the aging brain [162,215,312,313]. 

### 4.4. Non-Pharmacologic and Non-Invasive Strategies to Promote Neuroplasticity during Aging

As described above, molecular and structural changes within the brain with aging can contribute to a decline in brain function and neurodegeneration [314]. However, the cognitive reserve (i.e., the ability of the brain to cope with damage and deterioration) can significantly reduce the risk of dementia and other age-related neurodegenerative conditions. Various non-invasive and non-pharmacological approaches have been shown to increase the cognitive reserve and potentially counteract the deleterious effects of aging by protecting the brain against age-associated neurodegenerative processes [315]. These strategies, including physical exercise, environmental enrichment and social stimulation, a healthy diet, and caloric restriction, as well as sleep hygiene, have been shown to enhance brain plasticity and improve cognitive function in aging individuals [316] while also counteracting several age-induced alterations in brain signaling, structure, and function [63]. The following sections outline some of the non-invasive and non-pharmacological strategies proposed to increase neuroplasticity in the aging brain.

#### 4.4.1. Physical Exercise

Physical exercise is a well-established, non-invasive strategy for promoting neuroplasticity during aging [317,318]. Exercise is known to increase the production of growth factors, such as BDNF, which promote the survival and growth of neurons and synapses and play a key role in neuroplasticity [319]. In addition, exercise has also been shown to increase the expression of synapsin-I, a presynaptic protein related to motor performance [320].

Various studies have now shown the beneficial effects of exercise in maintaining cognitive function in aging [162,318,321,322]. Indeed, physical exercise increased hippocampal volume, improved spatial memory in older adults [321], and promoted cognitive function in individuals with mild cognitive impairment and dementia [322,323,324]. A systematic review and meta-analysis of 17 randomized controlled trials found that exercise interventions improved cognitive function, including memory, attention, processing speed, and executive function, in older adults [325]. A separate meta-analysis of 29 randomized controlled trials also found that exercise interventions were associated with significant improvements in cognitive function in healthy older adults [326]. A recent study using transcranial magnetic stimulation (TMS) found that older adults who engaged in regular physical exercise had more remarkable cortical plasticity than those who did not exercise regularly [327]. This suggests that exercise directly impacts the ability of the brain to adapt and change in response to environmental stimuli during aging. In addition, a study using magnetic resonance imaging (MRI) found that older adults who engaged in regular physical activity had greater gray matter volume in the prefrontal cortex, a brain region important for higher-order cognitive functions and decision making [328].

Together, these studies suggest that exercise can be a powerful tool for promoting neuroplasticity and cognitive health during aging. By increasing trophic support through the production of BDNF and promoting changes in brain structure and function, exercise may help protect the brain against age-related cognitive decline and improve the quality of life in later years. 

#### 4.4.2. Cognitive Stimulation and Socialization

Several lines of evidence have suggested that cognitive stimulation can effectively promote neuroplasticity and brain health during aging. Cognitive stimulation encompasses activities that challenge the brain, such as reading, writing, playing cognitive games, or learning new skills. These activities promote the formation of new neuronal connections and can help maintain cognitive function during aging [329]. A systematic review and meta-analysis found that engaging in mentally stimulating activities was associated with a lower risk of cognitive decline and dementia [330]. Older adults engaged in mentally stimulating activities had greater gray matter volume in brain regions important for memory and cognitive function [331], and an active cognitive lifestyle is associated with a more favorable cognitive trajectory in older persons [332]. Leisure activities such as reading, playing board games, playing musical instruments, and dancing were associated with a reduced risk of dementia in individuals older than 75 [333]. Learning new skills is one way to engage in mentally stimulating activities. In fact, learning a new skill, such as juggling, was associated with changes in brain structure and function, including increased gray matter volume in the visual and motor areas of the brain [81,82].

Socially interacting with others, such as friends and family, and engaging in social activities can also promote cognitive and emotional stimulation. Older adults who engaged in social activities, such as volunteering or participating in community events, had a lower risk of cognitive decline than those who did not engage in such activities [334]. On the other hand, a study found that social isolation was associated with a higher risk of dementia in older adults. This study followed over 2000 older adults for up to 7 years and found that those who were socially isolated had a 60% higher risk of developing dementia than those with social support [335]. Of note, socialization has been shown to promote the formation of new neuronal connections and enhance cognitive function. Indeed, socializing with others was associated with increased gray matter volume in brain regions important for memory and social cognition [336].

Furthermore, several studies showed that genetic and lifestyle factors play a role in determining the individual risk of dementia and cognitive impairment. The ε4 allele of the apolipoprotein E (APOE ε4) gene is the strongest known genetic risk factor for dementia and cognitive impairment. The association between APOE ε4 and faster cognitive decline was reduced in participants who were regularly engaged in productive activities [337].

#### 4.4.3. Diet and Caloric Restriction

Diet plays a significant role in promoting neuroplasticity, especially during the aging process. The brain is a complex organ that requires numerous nutrients, including vitamins, minerals, antioxidants, and essential fatty acids [338]. Several studies have found that a healthy diet rich in fruits, vegetables, whole grains, and lean protein provides the nutrients necessary for optimal brain function [339,340].

In particular, the role of omega-3 fatty acids in promoting neuroplasticity is well documented. These fatty acids are essential for brain health and are found in high concentrations in fatty fish such as salmon, sardines, and mackerel [341]. For example, a randomized controlled trial involving older adults with mild cognitive impairment found that supplementation with omega-3 fatty acids for 12 weeks significantly improved cognitive function compared to the placebo group [342]. In a different study, supplementation with omega-3 fatty acids was also shown to improve cognitive function in healthy older adults [343]. Furthermore, Dullemeijer et al. (2007) found that higher plasma levels of n-3 polyunsaturated fatty acids were associated with a reduced decline in sensorimotor speed and complex cognitive processing in older adults [344]. In addition to omega-3 fatty acids, macro and micronutrients present in balanced diets, such as B vitamins and flavonoids, can prevent or mitigate age-related degenerative processes [345].

Conversely, diets high in saturated and trans fats have been linked to cognitive decline and dementia. A systematic review and meta-analysis of prospective studies found that a higher intake of saturated fats was associated with a higher risk of cognitive impairment and dementia [346]. Similarly, a study by Morris et al. found that a diet high in saturated fat was associated with a greater risk of developing AD [347]. In agreement, Solfrizzi et al. (2006) found that a diet high in trans fats was associated with cognitive decline in older adults [348]. Of note, trans fats have been shown to have adverse effects on brain function, as they can impair synaptic plasticity, alter membrane composition, and increase neuroinflammation [346]. These findings suggest that diets high in saturated and trans fats may negatively impact brain health and should be avoided.

Caloric restriction is another dietary strategy that promotes neuroplasticity and improves cognitive function. Caloric restriction involves reducing calorie intake by a certain percentage while maintaining adequate nutrition [349]. Animal studies have demonstrated that caloric restriction can enhance neuroplasticity by increasing the production of neurotrophic factors, such as BDNF [350,351]. Recent human studies have also suggested that caloric restriction may benefit brain function. A randomized controlled trial of overweight adults found that a 25% reduction in calorie intake for two years resulted in significant improvements in verbal memory and executive function compared to the control group [352].

#### 4.4.4. Sleep Hygiene and Quality of Sleep

Sleep is critical in maintaining brain health and cognitive function, particularly in older adults. Numerous studies have demonstrated the importance of sleep in promoting neuroplasticity, particularly during learning and memory consolidation [353,354]. Sleep is also essential for clearing out toxic waste products and allowing the brain to regenerate [355].

Sleep deprivation has been associated with cognitive decline, particularly in older adults. Indeed, chronic sleep deprivation has been linked to a faster rate of cognitive decline in this population [356], while poor sleep quality was associated with a greater risk of developing dementia [357]. Recently, a study by Lucey and colleagues (2021) found that disrupted sleep was associated with increased levels of AD-related proteins in the cerebrospinal fluid [358].

Given this, and to promote optimal brain function, it is recommended that older adults sleep 7–8 h per night [359]. Also, maintaining good sleep hygiene practices, such as avoiding caffeine and electronic devices before bedtime and creating a relaxing sleep environment, can help improve the quality and quantity of sleep. Several lines of evidence have also highlighted the importance of regular exercise, healthy eating, and managing stress for promoting good sleep and overall brain health [360,361].

## 5. Conclusions

Neurodevelopmental exposures, numerous lifestyle factors, acute neurological processes (such as stroke and TBI), and neurodegenerative processes (such as AD and PD) can all disrupt neuroplasticity, leading to impairments in motor skills, affective behaviors, and cognitive function. Nevertheless, recent studies have highlighted the brain’s ability to compensate for these impairments through processes involving neural reorganization, which consists of recruiting other brain regions and neuronal circuits to compensate for the damaged ones [362]. Psychological traits, such as personality, motivation, and attention, also play a significant role in neuroplasticity mechanisms. For instance, individuals with high levels of motivation have been shown to exhibit greater neuroplasticity than those with low levels of motivation [363].

Understanding individual differences in experience-dependent neuroplasticity mechanisms is critical for developing personalized approaches to improving cognitive function and promoting recovery from brain injury or disease. In addition, promoting healthy lifestyles, including physical exercise, cognitive and social stimulation, a healthy and balanced diet, and good sleep hygiene, can go a long way in preventing or halting many age-related conditions and promoting overall brain health. By promoting healthy lifestyles and optimizing personalized treatment options that regulate and promote neuroplasticity, we can effectively harness the power of brain plasticity.

## Data Availability

Not applicable.

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
