# Peer review of "Exploring the Role of Neuroplasticity in Development, Aging, and Neurodegeneration"

_brainsci, 2023, doi:10.3390/brainsci13121610_

Round 1

Reviewer 1 Report

Comments and Suggestions for Authors

Marzola et al. investigated the role of neuroplasticity in development, aging, and neurodegeneration. This is an interesting review paper, suggesting neuroplasticity is a dynamic process that can be modulated and affected by various factors, including age, experience, and environmental conditions. The authors demonstrated understanding these factors can aid in developing effective strategies to harness the power of neuroplasticity and minimize its negative effects, leading to better treatments and outcomes for various neurological and neurodegenerative conditions. There are a few concerns to be addressed to further improve the manuscript.

1.     In the section of 3.1, DOHaD (developmental origins of health and disease) theory should be referred.

2.     In the section of 3.4, brain regions where adult neurogenesis occurs other than hippocampus, including hypothalamus should also be discussed. Because recent studies reported that neural stem cells are also present in the hypothalamus and have been shown to influence systemic aging.

3.     The theme of this review should also be discussed from a perspective sex difference.

Author Response

Reviewer #1

Comments and Suggestions for Authors

Marzola et al. investigated the role of neuroplasticity in development, aging, and neurodegeneration. This is an interesting review paper, suggesting neuroplasticity is a dynamic process that can be modulated and affected by various factors, including age, experience, and environmental conditions. The authors demonstrated understanding these factors can aid in developing effective strategies to harness the power of neuroplasticity and minimize its negative effects, leading to better treatments and outcomes for various neurological and neurodegenerative conditions. There are a few concerns to be addressed to further improve the manuscript.

  1. In the section of 3.1, DOHaD (developmental origins of health and disease) theory should be referred.

Response: We thank the reviewer for bringing this to our attention. We have addressed the concern by incorporating an additional paragraph into Section 3.1 of the manuscript, specifically focusing on the Developmental Origins of Health and Disease (DOHaD) theory. This new paragraph reads as follows:

The developmental origins of the health and disease hypothesis (DOHaD) propose that environmental exposure during early life (particularly during the prenatal period) can permanently influence the long-term development of disease. The initial studies addressing this relationship observed the association between gestational malnutrition and the phenotypes of the offspring, as well as the risk of developing metabolic diseases such as obesity, diabetes, and cardiovascular diseases later in life. In accordance with this, later studies have identified epigenetic modifications in fetal DNA as a response to environmental stimuli, which can permanently alter protein expression and phenotypes in the offspring. Of note, some of the environmental factors causing epigenetic modifications besides maternal nutrition include smoking, maternal stress, and infection (Wadhwa et al., 2009; Lacagnina, 2019).

  1. In the section of 3.4, brain regions where adult neurogenesis occurs other than hippocampus, including hypothalamus should also be discussed. Because recent studies reported that neural stem cells are also present in the hypothalamus and have been shown to influence systemic aging.

Response: We appreciate the insightful suggestion made by the reviewer. In response to your comment, we have expanded the discussion in Section 3.4 to also encompass other brain regions where adult neurogenesis occurs, including the hypothalamus. This addition to our manuscript aims to provide a brief exploration of the topic, considering recent studies highlighting the role of neural stem cells in the hypothalamus and their impact on systemic aging. This new discussion on hypothalamic neurogenesis reads as follows:

In addition to the hippocampus and the SVZ, additional brain regions have also emerged as sites where adult neurogenesis can take place (Jurkowski et al., 2020). Indeed, animal studies have delineated neurogenic loci encompassing the hypothalamus (Evans et al., 2002), striatum (Parent et al., 1995; Suzuki and Goldman, 2003; Shapiro et al., 2009), substantia nigra (SN; Cassidy et al., 2003), cerebral cortex (Magavi et al., 2000), and amygdala (Bernier et al., 2002).

Evidence indicates that the genesis of neurons in these newly identified neurogenic areas is attributable to the migration of NSPCs, typically originating from the SVZ (Bernier et al., 2002; Cao et al., 2002; Dayer et al., 2005; Inta et al., 2008; Shapiro et al., 2009; Huttner et al., 2014). Concurrently, some studies have challenged this notion and proposed the existence of endogenous pools of NSPCs within these regions, capable of local replication and integration into neuronal circuits (Parent et al., 1995; Zecevic and Rakic, 2001; Evans et al., 2002; Jhaveri et al., 2018).

The hypothalamus, one of the major regulatory centers in the brain, controls various homeostatic processes, and hypothalamic neural stem cells (htNSCs) have been shown to interfere with these processes. Indeed, the hypothalamic neurogenesis is thought to influence metabolism and fat storage, as evidenced by multiple studies on the impacts of a high-fat diet (HFD) in mice (Kokoeva, 2005; Lee et al., 2012, 2014; Li et al., 2012). Furthermore, neurogenesis within the hypothalamus is also thought to contribute to behavioral and sexual functions, as elucidated in studies by Bernstein et al. (1993), Fowler et al. (2002), and Cheng et al. (2004). Additionally, emerging research suggests that neurogenesis in the hypothalamus undergoes alterations during aging (Zhang et al., 2017), prompting investigations into the potential implications of age-related changes in hypothalamic neurogenesis on overall physiological homeostasis and cognitive functions (Plakkot et al., 2023).

  1. The theme of this review should also be discussed from a perspective sex difference.

Response: We appreciate the insightful suggestion made by the reviewer. In response to your comment, we have added a short section on Sex Hormones and Neuroplasticity (Section 3.7) to our manuscript. This section aims to provide a brief overview of the effect of sex hormones (particularly those produced locally in the hippocampus) on neuroplasticity. Although a more comprehensive overview on the role of sex hormones on neuroplasticity is beyond the scope of the present review, we hope the addition of this short section will help fill out the gap pointed by the Reviewer. For a more in-depth discussion of this topic, readers can also be directed to the various excellent review articles we’ve cited throughout this Section.

This new Section on Sex Hormones and Neuroplasticity reads as follows:

Sex hormones and neuroplasticity

Sex hormones are now known to have widespread actions in both the male and female brains, through mechanisms thought to involve both genomic and nongenomic receptors. Indeed, many neural and behavioral functions are affected by sex hormones such as estrogens, including mood, cognitive function, blood pressure regulation, motor coordination, pain, and opioid sensitivity (Marroco and McEwen, 2016). Moreover, sex-specific differences have been reported with regards to hippocampal-dependent cognition and neurogenesis, suggesting that sex hormones are involved in these processes. Indeed, estrogens have been shown to modulate certain forms of spatial and contextual memory, as well as different forms of neuroplasticity including neurogenesis, primarily in the adult female hippocampus (Duarte-Guterman et al., 2015, Trivino-Paredes et al., 2016). 

Peripheral sex steroid hormones, including estrogens, progesterone, testosterone, and other androgens are able to cross the blood-brain barrier and reach the brain. Furthermore, hippocampal neurons are capable of synthesizing sex steroids de novo from cholesterol, since neural cells express all the enzymes required for the synthesis of estradiol and testosterone, the end products of sex steroidogenesis (Hojo et al., 2008, 2004; Kato et al., 2013; Okamoto et al., 2012; Prange-Kiel et al., 2003; Camacho-Arroyo et al., 2020). Regarding 17β-estradiol (E2) in particular, its synthesis in hippocampal neurons is homeostatically controlled by Ca2+ transients and is regulated by the release of gonadotropin-releasing hormone (GnRH) (Fester and Rune, 2021). Indeed, release of GnRH from GnRH-positive neurons in the hippocampus is thought to regulate the local synthesis of sex steroids in a sex-dependent manner and thus contribute to the sexual differentiation of hippocampal neurons during the perinatal period (Fester and Rune, 2021, Brunne and Rune et al., 2022). Of note, a GnRH-induced increase in estradiol synthesis appears to provide a link between the hypothalamus and the hippocampus, and this may underlie, at least in part, estrous cyclicity of spine density in the female hippocampus (Kato et al., 2013; Fester and Rune, 2021). Furthermore, sex hormones can initiate gene transcription and activate signaling cascades by utilizing genomic and non-genomic (McEwen et al., 2012; Marroco and McEwen, 2016) mechanisms that play a key role in coordinating various physiological and pathological neuroplasticity-related events, such as formation or remodeling of dendritic spines, neurogenesis, synaptogenesis, and myelination modulation (Brann et al., 2022). For example, the study by Lu et al. in 2019, employing a forebrain-neuron-specific aromatase knock-out mouse model, provided compelling genetic evidence of the involvement of neuron-derived E2 in modulating AKT-ERK and CREB-BDNF signaling cascades. This study established that neuron-derived E2 is essential for normal expression of LTP and other forms of synaptic plasticity, as well as cognitive function in both male and female brains (Lu et al., 2019). However, it has also been proposed that whereas E2 appears to be essential to maintaining synaptic transmission and synaptic connectivity in the female hippocampus, dihydrotestosterone appears to be crucial for synaptic transmission and synaptic connectivity in the male hippocampus (Okamoto et al., 2012; Fester and Rune, 2021; Brunne and Rune et al., 2022). As the expression of sex hormones varies throughout the lifespan, its effects on neuroplasticity during distinct periods of development, adulthood, and aging, must also be considered. Along these lines, strategies aimed at restoring and/or maintaining normal hormone levels in the brain throughout the lifespan such as physical exercise are thought to be beneficial in promoting brain health in general and neuroplasticity in particular (Bettio et al., 2020).

Reviewer 2 Report

Comments and Suggestions for Authors

Patrícia Ramos Marzola, Thayza Melzer, Eloisa Pavesi, Joana M. Gil-Mohapel, Patricia S. Brocardo

EXPLORING THE ROLE OF NEUROPLASTICITY IN DEVELOPMENT, AGING, AND NEURODEGENERATION.

COMMENTS FOR THE AUTHOR:

1. The presented research is an original and important for neurobiology. Neuroplasticity plays a crucial role in developing and maintaining brain function, including learning and memory, as well as in recovery from brain injury and adaptation to environmental changes. Therefore, the relevance of the review topic is undeniable. The manuscript is included all parts which needs for the review: Abstract, Introduction, Basic Part, Conclusion, References.

2. The title clearly and precisely reflects the field of the manuscript.

3. Abstract is it really a summary, include key findings and have an appropriate length.

4. This review examines the history of the appearance of this term, discusses various representations of neuroplasticity. The review gives an idea of the mechanisms of neuroplasticity in development, synaptogenesis. An important part of the review is the analysis of neurogenesis in adult organisms. Next, we are talking about the formation of neuroplasticity processes at different stages of development, the mechanisms of neurodegeneration, protein aggregation are considered.

5. In conclusion, the problems of possible compensation of age-related disorders and a number of diseases due to the mechanisms of neuroplasticity are considered in detail.

6. Final comments.

The manuscript is fully consistent to the stated theme.

Author Response

Reviewer #2

Comments for Authors

  1. The presented research is an original and important for neurobiology. Neuroplasticity plays a crucial role in developing and maintaining brain function, including learning and memory, as well as in recovery from brain injury and adaptation to environmental changes. Therefore, the relevance of the review topic is undeniable. The manuscript is included all parts which needs for the review: Abstract, Introduction, Basic Part, Conclusion, References.
  2. The title clearly and precisely reflects the field of the manuscript.
  3. Abstract is it really a summary, include key findings and have an appropriate length.
  4. This review examines the history of the appearance of this term, discusses various representations of neuroplasticity. The review gives an idea of the mechanisms of neuroplasticity in development, synaptogenesis. An important part of the review is the analysis of neurogenesis in adult organisms. Next, we are talking about the formation of neuroplasticity processes at different stages of development, the mechanisms of neurodegeneration, protein aggregation are considered.
  5. In conclusion, the problems of possible compensation of age-related disorders and a number of diseases due to the mechanisms of neuroplasticity are considered in detail.
  6. Final comments.

The manuscript is fully consistent to the stated theme.

Response: We Thank the Reviewer for the positive comments and are happy that the Reviewer considered our work to be of high quality and worthy of publication.

Reviewer 3 Report

Comments and Suggestions for Authors

Marzola et al. – brain sciences:

The authors reviewed the available literature about neuroplasticity and evaluated its role during development, aging, and neurodevelopment. The authors wrote a well-organized review in which they summarized the available literature to explain the mechanisms of neuroplasticity. Then they developed different sections analyzing different kinds of neuroplasticity and how operates in both physiological and pathological conditions. The authors chose the proper language, the sentences are straightforward and they cited the more appropriate references. The topic is original and important. This review collected information about different kinds of neuroplasticity, they explained the mechanisms in a straightforward way that is very helpful to understand these mechanisms. They chose the proper and recent references and added important pieces of information about the involvement of these mechanisms in different pathological and physiological processes. 

Minor comments:

-check and fix references in the main text, sometimes are not numbered but appear as names and years (pages: 1, 4, 8, 10, 11, 12, 13, 15, 16, 17, and 18)

-references #170, 226, and 246, are mentioned only in the list of references but not in the main text

-on page 13, the title “4.2.2. Mitochondrial dysfunction and oxidative stress in neurodegeneration” has a different format compared to previous ones

-on page 14, the title “4.2.3. Neuroinflammation in neurodegeneration” has a different format compared to previous ones (italic). 

Author Response

Reviewer #3

Comments and Suggestions for Authors

Marzola et al. – brain sciences:

The authors reviewed the available literature about neuroplasticity and evaluated its role during development, aging, and neurodevelopment. The authors wrote a well-organized review in which they summarized the available literature to explain the mechanisms of neuroplasticity. Then they developed different sections analyzing different kinds of neuroplasticity and how operates in both physiological and pathological conditions. The authors chose the proper language, the sentences are straightforward and they cited the more appropriate references. The topic is original and important. This review collected information about different kinds of neuroplasticity, they explained the mechanisms in a straightforward way that is very helpful to understand these mechanisms. They chose the proper and recent references and added important pieces of information about the involvement of these mechanisms in different pathological and physiological processes.

Response: We Thank the Reviewer for the positive comments and are happy that the Reviewer considered our work to be of high quality and worthy of publication.

Minor comments:

- check and fix references in the main text, sometimes are not numbered but appear as names and years (pages: 1, 4, 8, 10, 11, 12, 13, 15, 16, 17, and 18)

Response: We thank the reviewer for pointing out these errors in the formatting of some of our References. The have been fixed in the revised version of our manuscript.

- references #170, 226, and 246, are mentioned only in the list of references but not in the main text

Response: We thank the reviewer for pointing out these errors in the citing of some of our References. The have been fixed in the revised version of our manuscript.

- on page 13, the title “4.2.2. Mitochondrial dysfunction and oxidative stress in neurodegeneration” has a different format compared to previous ones.

Response: We thank the reviewer for pointing out this inconsistency in the format of this section’s title. This has been fixed in the revised version of our manuscript.

 - on page 14, the title “4.2.3. Neuroinflammation in neurodegeneration” has a different format compared to previous ones (italic).

Response: We thank the reviewer for pointing out this inconsistency in the format of this section’s title. This has been fixed in the revised version of our manuscript.
